

# Recent jet and jet substructure measurements at the LHC, and ML based tagging

**Meena.Meena$^\star$ on behalf of the ATLAS and CMS Collaborations**

Panjab University, Chandigarh, India

$\star$ [meena.meena@cern.ch](mailto:meena.meena@cern.ch)

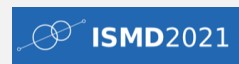

*50th International Symposium on Multiparticle Dynamics*
*(ISMD2021)*
*12-16 July 2021*
doi:[10.21468/SciPostPhysProc.10](https://doi.org/10.21468/SciPostPhysProc.10)

## Abstract

Recent jet and jet substructure measurements at the LHC, and of machine-learning-based tagging techniques are presented using proton-proton collision data collected by the ATLAS and CMS experiments at CERN's Large Hadron Collider. These measurements are crucial for precise tests of electroweak and pQCD calculations and searches for physics beyond the Standard Model. The measurements are compared with several Monte Carlo event generators predictions which provide valuable input to the tuning of perturbative and non-perturbative models and to constraining model parameters of advanced parton-shower Monte Carlo programs.

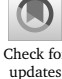
doi:[10.21468/SciPostPhysProc.10.022](https://doi.org/10.21468/SciPostPhysProc.10.022)

## 1 Introduction

In this document, an overview of recent jet and jet substructure measurements at the LHC, and of machine-learning (ML) based tagging techniques are presented. Analyses are performed using proton-proton (pp) collision data recorded at the ATLAS [1] and CMS [2] experiments at a center-of-mass-energy ($\sqrt{s}$) = 13 TeV. Comparisons of the results with various theoretical predictions are presented.

## 2 Measurement of the Lund jet plane using charged particles

A double-differential cross-section measurement of the Lund jet plane of primary jet emissions is performed using data collected with the ATLAS detector corresponding to an integrated luminosity ($L_{int}$) of 139 fb$^{-1}$ [3]. A two-dimensional space spanned by $\ln(1/z)$ and $\ln(1/\theta)$, where z is the momentum fraction of the emitted gluon relative to the primary quark or gluon core and $\theta$ is the emission opening angle. This space is called the Lund plane (LP). The

LP is not an observable but it can be approximated by using the softer (harder) proto-jet to represent the emission (core) in the original theoretical depiction. For each proto-jet pair, at each declustering step of jets formed using the Cambridge/Aachen algorithm, an entry is made in the approximate LP (henceforth, the 'primary Lund jet plane' or LJP) using the observables $\ln(1/z)$ and $\ln(R/\Delta R)$, R is the jet radius parameter, and $\Delta R$ measures the angular separation. Using this grooming procedure, individual jets are represented as a set of points within the LJP. The schematic representation of the LJP is shown in Figure 1 (left). It is observed that varying the default angle-ordered to dipole parton-shower (PS) model in Herwig 7.1.3 (hadronization model in Sherpa 2.1.1) results in differences of up to 50% in the perturbative hard and wide-angle emissions (softer and more collinear emissions at the boundary between perturbative and non-perturbative regions). Varying the matrix element (ME) from leading order (LO) in Pythia 8.230 to next-to-leading order (NLO) in Powheg+Pythia 8.230 causes small changes of up to 10% in the region populated by the hardest and widest-angle emissions.

The measurement is performed on an inclusive selection of dijet events, with a leading jet transverse momentum $(p_T) > 675$ GeV and pseudorapidity $(|\eta|) < 2.1$. Particle-level charged hadrons and their reconstructed tracks are also used. Tracks are required to have $p_T > 500$ MeV and must be associated with the primary vertex with the largest sum of track $p_T^2$ in the event. The average number of emissions in the fiducial region is measured to be $7.34 \pm 0.03$ (syst.) $\pm 0.11$ (stat.). The data are compared with predictions from several Monte Carlo (MC) generators for four selected horizontal and vertical slices through the LJP. Figure 1 (middle) shows the LJP region, where emissions change from wide-angled to more collinear, the distribution passes through a region sensitive to the choice of PS model, and then enters a region which is instead sensitive to the hadronization model. The differences between PS (hadronization) algorithms implemented in Herwig 7.1.3 (Sherpa 2.2.5) are notable at large (small) values of $k_T$, where the two models disagree most significantly for hard emissions reconstructed at the widest angles (soft-collinear splittings at the transition between perturbative and non-perturbative regions of the plane). The Powheg+Pythia and Pythia predictions only differ significantly for hard and wide-angle perturbative emissions, where ME corrections are relevant. No prediction describes the data accurately in all regions, the Herwig 7.1.3 angle-ordered prediction provides the best description across most of the plane. This measurement illustrates the ability of the Lund jet plane to isolate various physical effects, and will provide useful input to both perturbative and non-perturbative model development and tuning.

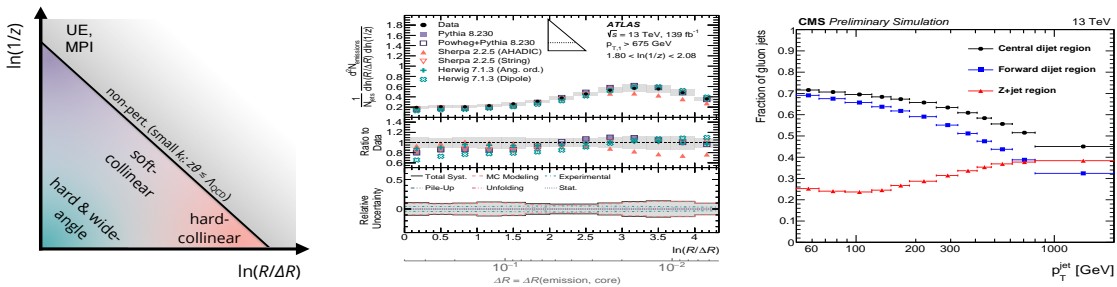

Figure 1: Schematic representation of the LJP (left). Unfolded LJP for $1.80 < \ln(1/z) < 2.08$ (middle) [3]. Fraction of AK4 gluon jets in the Z+jet region, and the central and forward jets in the dijet region (right) [4].

# 3 Study of quark and gluon jet substructure in dijet and Z+jet events

A measurement of jet substructure observables describing the distribution of particles within quark- and gluon-initiated jets is performed with the CMS detector corresponding to $L_{int}$ of 35.9 fb $^{-1}$ [4]. The fiducial phase space regions are defined by selecting Z boson with atleast one inclusive jet in the Z+jet event sample and the atleast two jets in the dijet event sample at reconstructed (generator)-level jets with $p_T >$ 30 (15) GeV and rapidity ($|y|$) < 2.4. Figure 1 (right) shows that Z+jet sample is dominated by 64–76% quark jets and central and forward dijet jets are dominated by 69–72% (55–68% ) gluon (quark) jets at low (high) $p_T$. In this analysis, measurements of a set of five observables $\lambda_\beta^\kappa$ are also reported that distinguish quark- and gluon-initiated jets. The generalized angularities are defined as:

$$\lambda_\beta^\kappa = \sum_{i \in jet} \left[ z_i^\kappa \frac{(\Delta R_i)}{R})^\beta \right], \tag{1}$$

where $z_i$ is the fractional transverse momentum carried by the i$^{th}$ jet constituent, R is the jet size parameter, and $\Delta R_i$ is the displacement of the constituent from the jet axis. The five observables are Les Houches angularity (LHA) ($\lambda_{0.5}^1$), width ($\lambda_1^1$), thrust ($\lambda_2^1$), multiplicity ($\lambda_0^0$), and $(p_T^D)^2$ ($\lambda_0^2$). The LHA, width, and thrust are infrared and collinear safe and particularly sensitive to the modelling of perturbative emissions in jets, while the other two have larger contributions from non-perturbative effects. Experimental data distributions unfolded to the particle-level. The mean of LHA, width, and thrust distributions decreases as a function of the jet $p_T$ in both the Z+jet and central dijet regions, as a result of constituents being located closer to the jet axis due to the larger Lorentz boost at higher $p_T$. This trend is displayed by all generators. In the gluon-enriched sample, HERWIG7 CH3 [5] – [6], PYTHIA8 CP5 [7], PYTHIA8 CP2 [7], and SHERPA [8] generally provide a better description than either HERWIG++ [9] – [10] or MG5+PYTHIA8 [11] – [12]. In the quark-enriched sample, MG5+PYTHIA8 provides the best description, followed by HERWIG7, PYTHIA8 CP2, SHERPA, HERWIG++, and PYTHIA8 CP5. Improved modelling of gluon jets at the cost of poorer modelling of quark jets is observed. The ratio of the means in the central dijet and Z+jet region are also compared. There is similar data-to-simulation agreement for AK8 versus AK4 jets, charged-only versus charged+neutral and groomed versus ungroomed observables. At low $p_T$, the ratio is found to be significantly larger than unity for LHA, width, thrust, and multiplicity, and significantly smaller for $(p_T^D)^2$. It indicates that these observables have significant separation power between quark and gluon jets and clear need for improvements in the MC. The best overall data-to-simulation agreement for the ratio is achieved by SHERPA, followed by HERWIG++, MG5+PYTHIA8, HERWIG7 CH3, PYTHIA8 CP2, and PYTHIA8 CP5.

# 4 Mass regression of highly-boosted jets using graph neural networks

A novel technique based on machine learning for reconstructing the true mass in hadronic decays of highly Lorentz-boosted top quarks and W, Z, and Higgs bosons [13] is presented. The technique, commonly known as mass regression, is based on ParticleNet [14] – [16], a graph neural network using an unordered set of jet constituent particles as the input. In ParticleNet Mass Regression, the training sample consists of an equal mix of QCD and Higgs bosons events, generated with MadGraph5 (hard scattering) and Pythia8 (PS and hadronisation) with 2018 data conditions. The Higgs boson sample has been generated with an equal mix of H $\rightarrow$

bb/cc/qq (q=u,d,s) decays and its MC mass has been taken from a uniform distribution in the [15, 250] GeV range. The groomed jet mass obtained from the "modified mass drop tagger" algorithm, also known as the "soft drop" algorithm [17] – [19], with angular exponent $\beta = 0$, soft cutoff threshold $z_{cut} = 0.1$, is used to remove soft, wide-angle radiation from the jet. The soft drop mass is calibrated in a top quark-antiquark sample enriched in hadronically decaying W bosons. The target mass is defined as the "soft drop" mass of the associated truth particle-level jet for the QCD sample and the Higgs boson generator mass ($m_H \in$ [15, 250] GeV) for the Higgs boson sample respectively. The mass response for large-R (R=0.8) Higgs boson jets with $p_T > 400$ GeV and $100 < M_{target} < 150$ GeV for H → bb jet compositions is shown in Figure 2 (left). The resolution degrades for the heavier quark flavors due to the larger presence of neutrinos. In addition, there is significant improvement in tails with the mass regression, in particular at M≈0, where the soft drop algorithm incorrectly identifies the large R jet as single-prong. The effective resolution ($\sigma_{eff}$(m)/m) is computed as half of the minimum interval containing the mode and 68% of the area under the response distributions (defined as $M_{reco}/M_{target}$). This definition provides a robust estimate of the resolution in particular for distribution with large skewness and fat-tails. The mass regression for H → cc jet compositions shows a substantial improvement in the mass resolution and in the absolute scale as compared to the more traditional grooming algorithms such as soft drop for all the jet compositions AK8/15.

## 5  Boosted hadronic vector boson and top quark tagging

The latest development and optimization of taggers to identify high-$p_T$ (boosted) hadronic decays of W and Z boson and top quarks, as well as their calibration is performed [20] using data collected by the ATLAS experiment between 2015 and 2017 that correspond to $L_{int}$=80 fb$^{-1}$. W and Z boson taggers are defined using selection criteria on individual hadronic jet properties (cut-based algorithms) such as large-R jet mass, energy correlation function ratio ($D_2^{\beta=1.0}$) [21], and the ghost-associated track multiplicity ($n_{track}$). Top quark taggers are defined using deep neural networks that use jet substructure moments as input. The W and Z taggers are evaluated in MC simulation and data for jets with 200 GeV $< p_T <$ 2.5 TeV and the top tagger for jets with 350 GeV $< p_T <$ 4 TeV. The additional cut on $n_{track}$ has been found to improve background rejection for a fixed signal efficiency due to the rejection of jets seeded by gluons. In the case of the signal efficiency, the scale factors are derived using $t\bar{t}$ events in the muon+jets channel while $\gamma$+jet and multijet events are used to derive scale factors for quark and gluon-initiated background jets covering the $p_T$ ranges [200, 2000] GeV and [500, 3500] GeV, respectively. The tagger signal efficiencies are extracted from distributions of the leading large-R jet $m_{comb}$ from candidate $t\bar{t}$ events that are partitioned in $p_T$ bins covering the ranges [350, 1000] GeV for top tagging and [200, 600] GeV for W tagging. The $m_{comb}$ is the weighted sum of masses from calorimeter information and from the mixed calorimeter and tracking information. The deep neural-network scores for inclusive top taggers are shown in Figure 2 (middle) for events that pass the top selection in data and MC. There is good agreement between data and MC across the region within the uncertainties considered. Overall it is observed that the signal efficiency scale factors range between 0.8 and unity and background efficiency scale factors are close to unity for all the taggers considered. Figure 2 (right) shows signal efficiency scale factors the for the inclusive top taggers. A method of extrapolating scale factors and uncertainties using only MC is employed to extend the range of validity to $p_T \leq$ 2.5 TeV for the W and Z taggers and to $p_T \leq$ 4 TeV for the top taggers. The reconstruction and identification of boosted objects provide powerful handles for new physics searches as well as precision measurements of SM processes.

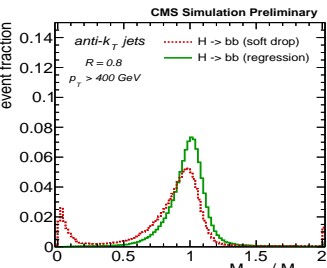
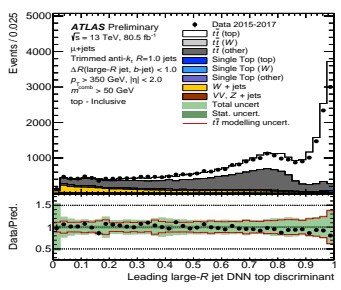
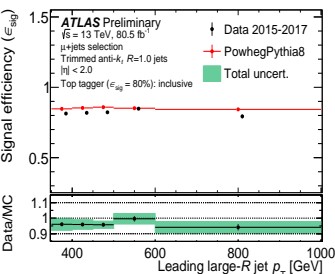

Figure 2: Performance of the ParticleNet regression (solid) and the soft drop algorithm (dashed) [13]. The mass response is shown for $100 < M_{target} < 150$ GeV for H → bb jet compositions (left). Distribution of the DNN score for the leading large-R jets in top-enhanced single-muon $t\bar{t}$ events (middle). The measured efficiencies for the 80% for inclusive top taggers (right) [20].

## 6 Conclusion

The ATLAS and CMS experiments have a rich program of measurements related to jet and their substructure. Here, we presented an overview of several recent measurements which are sensitive to different theoretical approaches and useful to constrain the model parameters of advanced Monte Carlo programs.

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
