# Peer review of "Recent jet and jet substructure measurements at the LHC, and ML based tagging"

_SciPost Physics Proceedings, doi:SciPost Phys. Proc. 10, 022 (2022)_

## Round 1 · Referee Report · Anonymous (Referee 1) · 2022-1-10

Report

This is a proceeding reporting on published ATLAS+CMS jet substructure results. While I understand it is a challenge to include and explain such a diverse set of measurements, I do find the presentation needs some improvement. I tried to point out the ones I could see, but I would strongly suggest the author to go over the text carefully.

  1. Abstract: Recent results are presented of jet and jet substructure measurements at the LHC --> Recent jet and jet substructure measurements at the LHC are presented ...

  2. Abstract: I am not sure electroweak is relevant for these results?

  3. The measurements are compared with several Monte Carlo event generators -> generator predictions

  4. tuning of perturbative and non-perturbative models -> we rarely tune the perturbative part, do we?

  5. The sentence in Introduction is too long, and not complete.

  6. I think LJP missed mentioning that its using tracks.

  7. Figure1. caption overlaps with axis labels

  8. Sec 3, use an mbox to not split a formula, space in at least,

  9. Sec3, Z+jet sample dominated -> is dominated

  10. Sec3, ith

  11. Mixture of particle and generator level

  12. Generator+tunes need ref

  13. Sec 4, The groomed jet mass obtained from ... -> this sentence needs rephrasing

  14. Sec 4, larger presence of neutrinos -> rephrase

16.Sec 4, tails are strongly mitigated-> not clear

  1. Can we avoid appendix here?

  2. Sec 5, first line, rephrase

  3. Sec 5, data is collected -> are

  4. Sec 5, deep neural networks (advance Machine Learning techniques) -> I am not sure the text in bracket is needed!

  5. Sec 5, The addition cut -> additional

Please use FloatBarrier appropriately to bring figures closer to text, and certainly not lumped at the end after references.

Please check formatting, spaces before section headings seem very inconsistent and squeezed.

  • validity: -
  • significance: -
  • originality: -
  • clarity: -
  • formatting: -
  • grammar: -

Author:  Meena Meena  on 2022-01-31  [id 2140]

(in reply to Report 1 on 2022-01-10)
Category:
answer to question

  1. Abstract: Recent results are presented of jet and jet substructure measurements at the LHC --> Recent jet and jet substructure measurements at the LHC are presented ...

Author: Text are modified as follows: Recent jet and jet substructure measurements at the LHC, and of machine-learning-based tagging techniques are presented using proton-proton collision data collected by the ATLAS and CMS experiments at CERN’s Large Hadron Collider.

  1. Abstract: I am not sure electroweak is relevant for these results?

Author: Electroweak is relevant here because the fundamental parameters of the electroweak theory relate the masses and transverse momentum, etc. of the W and Z bosons, via the electroweak mixing angle. The precise knowledge of these allows us to make a precise prediction for SM and BSM.

  1. The measurements are compared with several Monte Carlo event generators -> generator predictions

Author: Done

  1. tuning of perturbative and non-perturbative models -> we rarely tune the perturbative part, do we?

Author: yes

  1. The sentence in Introduction is too long, and not complete.

Author: I modify the text as follow: In this document, an overview of recent jet and jet substructure measurements at the LHC, and of machine-learning (ML) based tagging techniques are presented. Analyses are performed using ....

  1. I think LJP missed mentioning that its using tracks.

Author: I wish to write more but there is limitation of pages.

  1. Figure1. caption overlaps with axis labels

Author: Fixed

  1. Sec 3, use an mbox to not split a formula, space in at least,

Author: Could you please elaborate more? The equation format seems ok for me.

  1. Sec3, Z+jet sample dominated -> is dominated

Author: Fixed

  1. Sec3, ith

Author: Fixed

  1. Mixture of particle and generator level

Author: Could you please specify the section or text of that line?

  1. Generator+tunes need ref

Author: I wish to write more but there is limitation of pages.

  1. Sec 4, The groomed jet mass obtained from ... -> this sentence needs rephrasing

Author: It seems fine to me. Could you please suggest some text?

  1. Sec 4, larger presence of neutrinos -> rephrase

Author: It seems fine to me. Could you please suggest some text?

16.Sec 4, tails are strongly mitigated-> not clear

Author: I meant that at M≈0 where the soft drop algorithm incorrectly identifies the large R jet as a single jet but it could be 2 or more jets.

  1. Can we avoid appendix here?

Author: Done

  1. Sec 5, first line, rephrase

Author: Done

  1. Sec 5, data is collected -> are

Author: The text is modified as follow: "using data collected by the ATLAS experiment between 2015 and 2017 that"

  1. Sec 5, deep neural networks (advance Machine Learning techniques) -> I am not sure the text in bracket is needed!

Author: ok. Text in bracket is removed.

  1. Sec 5, The addition cut -> additional

Author: Done

---

## Editorial Decision

published